# The Cannabinoid Pharmacology of Bone Healing: Developments in Fusion Medicine

**DOI:** 10.3390/biomedicines13081891

**Published:** 2025-08-03

**Authors:** Gabriel Urreola, Michael Le, Alan Harris, Jose A. Castillo, Augustine M. Saiz, Hania Shahzad, Allan R. Martin, Kee D. Kim, Safdar Khan, Richard Price

**Affiliations:** 1Department of Neurological Surgery, School of Medicine, University of California Davis, 4610 X Street, Sacramento, CA 95817, USA; miqle@ucdavis.edu (M.L.); alan.harris@louisville.edu (A.H.); jcastillo@ucdavis.edu (J.A.C.J.); armartin@health.ucdavis.edu (A.R.M.); kdkim@health.ucdavis.edu (K.D.K.); 2School of Medicine, University of Louisville, 500 S Preston Street, Louisville, KY 40202, USA; 3Department of Orthopaedic Surgery, School of Medicine, University of California Davis, 4860 Y Street, Sacramento, CA 95817, USA; amsaiz@ucdavis.edu (A.M.S.); hshahzad@ucdavis.edu (H.S.); safkhan@ucdavis.edu (S.K.)

**Keywords:** endocannabinoid system, spinal fusion, pseudarthrosis, CB2 agonist, cannabidiol, tetrahydrocannabinol, bone-mineral density

## Abstract

**Background/Objectives:** Cannabinoid use is rising among patients undergoing spinal fusion, yet its influence on bone healing is poorly defined. The endocannabinoid system (ECS)—through cannabinoid receptors 1 (CB1) and 2 (CB2)—modulates skeletal metabolism. We reviewed preclinical, mechanistic and clinical evidence to clarify how individual cannabinoids affect fracture repair and spinal arthrodesis. **Methods:** PubMed, Web of Science and Scopus were searched from inception to 31 May 2025 with the terms “cannabinoid”, “CB1”, “CB2”, “spinal fusion”, “fracture”, “osteoblast” and “osteoclast”. Animal studies, in vitro experiments and clinical reports that reported bone outcomes were eligible. **Results:** CB2 signaling was uniformly osteogenic. CB2-knockout mice developed high-turnover osteoporosis, whereas CB2 agonists (HU-308, JWH-133, HU-433, JWH-015) restored trabecular volume, enhanced osteoblast activity and strengthened fracture callus. Cannabidiol (CBD), a non-psychoactive phytocannabinoid with CB2 bias, accelerated early posterolateral fusion in rats and reduced the RANKL/OPG ratio without compromising final union. In contrast, sustained or high-dose Δ^9^-tetrahydrocannabinol (THC) activation of CB1 slowed chondrocyte hypertrophy, decreased mesenchymal-stromal-cell mineralization and correlated clinically with 6–10% lower bone-mineral density and a 1.8–3.6-fold higher pseudarthrosis or revision risk. Short-course or low-dose THC appeared skeletal neutral. Responses varied with sex, age and genetic background; no prospective trials defined safe perioperative dosing thresholds. **Conclusions:** CB2 activation and CBD consistently favor bone repair, whereas chronic high-THC exposure poses a modifiable risk for nonunion in spine surgery. Prospective, receptor-specific trials stratified by THC/CBD ratio, patient sex and ECS genotype are needed to establish evidence-based cannabinoid use in spinal fusion.

## 1. Introduction

Cannabis has been used medicinally for millennia, with archaeological evidence of its use dating back over 6000 years in Central Asia [1]. Ancient Chinese records (~2700 BC) describe cannabis preparations for pain relief and as surgical anesthesia [2,3]. By ~1000 BC, medical use of cannabis had spread to India, where it was employed for analgesia, anticonvulsant therapy, and other remedies [4]. Cannabis later entered Western medicine in the 19th century notably through W.B. O’Shaughnessy’s 1843 reports on its efficacy for pain and muscle spasms [5]. These historical observations underscore the longstanding interest in cannabis’s therapeutic potential (particularly for pain management), which frames modern investigations into its role in bone healing [6]. 

Although the medicinal roots of cannabis are grounded in China, its cultural and therapeutic use expanded more extensively in ancient India [4]. There, cannabis was not only used for medical purposes but also consumed recreationally. The Atharva Veda, one of India’s ancient religious texts, refers to cannabis as one of the five sacred plants, highlighting its psychoactive properties as a function of preparation [1]. By approximately 1000 BC, Indian physicians employed cannabis for a wide array of therapeutic purposes, including analgesia, anticonvulsant therapy, hypnosis, antimicrobial treatments, antispasmodic effects, appetite stimulation, digestive aid, and cough suppression [4]. From its Asian origins, cannabis spread across ancient Syria, Europe, and Africa, eventually reaching the Americas [3]. The formal introduction of cannabis into Western medicine occurred in the late 19th and early 20th centuries as European physicians most notably O’Shaughnessy, began documenting the clinical utility of the plant [5]. Understanding this historical backdrop is essential for contextualizing current and future investigations into the medical potential of cannabis.

Today, approximately 4% of the global adult population reports cannabis use, though prevalence varies by region: around 34% of Australians aged 14 and older, 40% of North Americans aged 12 and older, and 20% of Europeans aged 15 to 64 report lifetime exposure [7]. Despite ongoing debates over cannabis’s safety, a growing number of U.S. states and countries worldwide are moving toward legalization for medicinal and/or recreational use. In the context of modern Western medicine, where pharmaceutical and surgical interventions dominate, understanding cannabis’s effects on human health is increasingly important.

Over 100 cannabinoids have been isolated from Cannabis sativa, yet it was not until the early 1960s that cannabidiol (CBD) was first isolated in 1963, followed by the identification of THC in 1964 [8,9]. The discovery of the ECS and its receptors, primarily the cannabinoid receptor type 1 (CB1, encoded by the CNR1 gene) and type 2 (CB2, encoded by CNR2), marked a pivotal advancement, spurring a wave of mechanistic and physiological research across organ systems including the central nervous, gastrointestinal, and skeletal systems [10]. However, the specific effects of cannabis on bone physiology, and particularly on bone healing, remain poorly characterized. Existing evidence is conflicting: while some studies suggest ECS activation may enhance bone regeneration [11], others imply inhibitory effects [12]. Given the plant’s complex biochemical profile comprising multiple cannabinoids with varying receptor targets, and the context-dependent physiological responses they elicit, a clearer and more nuanced understanding is urgently needed.

Each year, over 10 million orthopedic and 1.2 million spinal surgeries are performed, with a growing proportion of patients reporting cannabis use in the preoperative and postoperative periods [13,14]. Considering this trend, the present manuscript aims to provide a comprehensive review of cannabis’s intersection with bone healing and spinal fusion. We seek to synthesize epidemiological findings, evaluate current clinical evidence, explore relevant molecular mechanisms and animal model data, and discuss how this emerging knowledge can inform future research directions and clinical decision-making.

## 2. Epidemiology

Cannabis use has become increasingly prevalent worldwide, driven by legalization for both medical and recreational purposes. According to the 2024 United Nations World Drug Report, approximately 228 million individuals reported cannabis use globally, surpassing other substances such as opioids (60 million users) and amphetamines (30 million users) [15]. In the United States, cannabis is now the most used recreational drug, with 19% of the general population reporting use in 2021 [16]. As of February 2025, 39 states and four U.S. territories permit medical cannabis use, while 24 states have legalized its recreational use for adults [17]. Legalization, along with aggressive marketing of cannabis’s health benefits, has contributed to growing public acceptance and a decline in perceived risks. One prior study reported a reduction in hospital admissions for chronic pain following legalization in Colorado, the first U.S. state to permit recreational use [12]. Nonetheless, the clinical utility of cannabis remains limited by the psychoactive and potentially addictive properties of its THC component.

Epidemiological data suggests a correlation between increased medical cannabis use and reduced opioid prescriptions. One study found that 61% of medical cannabis users cited pain relief as their primary reason for use [18]. Among patients undergoing total joint arthroplasty, 24% reported prior or current cannabis use, with 2.9% using it within one year of surgery. In a recent survey, 75% of patients who had previously undergone joint replacement indicated a willingness to consider cannabis for pain management if prescribed by a physician, and 39% supported its legalization for recreational purposes [19]. In the context of the ongoing opioid epidemic, cannabis is increasingly viewed as a potential alternative. Notably, cannabis use among adults aged 65 years and older has increased sixfold over the past decade, according to the American Association of Hip and Knee Surgeons [19]. Chronic musculoskeletal pain, arthritis, and insomnia are among the most frequently cited reasons for use in this age group [20].

These demographic trends intersect with spinal surgery, a field already experiencing high procedural volumes among older adults with degenerative spinal conditions. Consequently, an increasing number of spine surgeries are being performed in patients who use cannabis. Despite this, few studies have examined the perioperative implications of cannabis use in this population. As a result, surgeons currently lack evidence-based guidance on critical questions—such as whether cannabis modifies analgesic needs, influences postoperative opioid consumption, or affects bone healing and fusion biology. Given these knowledge gaps, there is an urgent need to investigate cannabis use within spine surgery cohorts to determine its impact on outcomes and identify any procedure-specific risks or benefits.

## 3. Current Clinical Evidence in Spine Surgery

Chronic back pain remains a challenging condition to manage, largely because most conventional analgesics are unsuitable for long-term use. Preliminary data, however, suggest a potential role for medicinal cannabis [20]. Price et al., demonstrate in a systematic review, incorporating four randomized control trials or prospective cohort studies that cannabis was shown to be effective in treating back pain with no documented adverse effects [21]. In a 2006 prospective cohort study, Pinsger et al. administered nabilone, a synthetic cannabinoid, to 30 patients with refractory degenerative back pain and observed significant reductions in pain scores [22]. Similarly, Yassin et al. reported improvements in both pain levels and functional outcomes among 31 patients with fibromyalgia-related back pain who received adjunctive medical cannabis (under physician supervision) alongside standard analgesic therapies [22]. While these findings are encouraging, both studies were limited by small sample sizes, non-randomized designs, and short follow-up periods of six months, insufficient to assess long-term efficacy, tolerance, or adverse effects.

Notably, these retrospective analyses predominantly reflect recreational cannabis use (i.e., patient self-use without medical oversight, often characterized as cannabis use disorder), in contrast to the regulated medical use in the studies above. Conversely, chronic cannabis use may negatively impact postoperative outcomes. Several studies have shown that cannabis users exhibit increased opioid consumption both during hospitalization and in the months following posterior lumbar fusion [23]. A multi-institutional analysis further revealed longer hospital stays and an elevated risk of future revision surgery among chronic cannabis users [24]. Similarly, database studies focusing on [25] patients with cannabis use disorder reported prolonged in-hospital lengths of stay (LOS) following one- or two-level lumbar fusions [24].

A prior meta-analysis of 788 patients across four studies found that cannabis users were 3.6 times more likely to require revision for pseudarthrosis after cervical fusion, though no significant association was observed for lumbar fusion [26]. A separate multi-institutional study reported a twofold increase in radiographic pseudarthrosis and revision within the first postoperative year among cannabis users [27]. Additionally, Lambrechts et al. demonstrated that preoperative marijuana use was associated with a higher risk of cervical spine reoperation, along with worse one-year postoperative outcomes on the Physical Component Score, Neck Disability Index, and Visual Analogue Scale for both arm and neck pain [28]. A PearlDiver database study of 40,989 lumbar spine fusions conducted between 2010 and 2020 identified a negative synergy between cannabis and tobacco use [28]. Pseudarthrosis risk was highest in patients who used both cannabis and tobacco; after adjusting for tobacco, cannabis use alone still carried a higher risk than no cannabis use. Taken together, the evidence indicates that perioperative cannabis use is independently associated with higher rates of pseudarthrosis and subsequent revision surgery, outcomes that can prolong hospitalization and escalate healthcare utilization. Although human evidence linking cannabinoids to bone mineral density (BMD) and turnover is limited, several small clinical studies have begun to probe this relationship. Case series provided preliminary but inconclusive findings following oral administration of CBD in two postmenopausal women with osteopenia. Participants were randomized to receive either 100 mg or 300 mg of CBD daily. The treatment was well tolerated and did not adversely affect sleep, mood, or quality of life. Notably, reductions in bone turnover markers were observed, including decreases in carboxyl-terminal collagen crosslinks (CTx) and markers of bone formation such as procollagen type 1 N-terminal propeptide (P1NP), bone-specific alkaline phosphatase (BSAP), and osteocalcin (OC) [29]. In a separate interventional study—the first to examine human biomarkers of bone turnover—short-term treatment with CBD- or THC-dominant medical cannabis attenuated markers of bone resorption, although the findings were not clinically significant. Importantly, this study did not evaluate dose-response effects within treatment groups [30]. An oromucosal spray formulation containing both THC and CBD has been approved in the United Kingdom for the treatment of spasticity, neuropathic pain, and multiple sclerosis-related symptoms, though it is not currently approved for use in the United States [31].

Overall, human clinical data on cannabis use in orthopedic and spinal care remains mixed. Studies evaluating back pain and spasticity have generally reported positive outcomes, with minimal adverse events or complications. However, despite its therapeutic potential, some reports suggest an increased risk of opioid misuse among cannabis users. Regarding bone healing, most available studies are retrospective and indicate that cannabis may inhibit bony union and reduce fusion rates. This trend contrasts with findings from in vivo and in vitro models, which often show neutral or even beneficial effects. Notably, in vitro cell culture studies and in vivo animal models often report neutral or positive effects of cannabinoids on bone (e.g., enhanced bone formation in mice), whereas retrospective clinical studies in humans have linked cannabis use to inhibited bone union. These conflicting outcomes can be partly explained by differences in dosage and context: cannabinoids exhibit biphasic effects (beneficial at one dosage and inhibitory at a higher dosage), and animal studies typically use controlled timing and doses that may not mirror the chronic, varied use patterns in patients. Additionally, species-specific responses and the presence of confounders (like nicotine co-use or metabolic differences) in human populations may mask or modify the direct effects observed in simplified models. Importantly, many clinical studies fail to account for key confounding factors—such as the specific cannabis strain used, the relative cannabinoid composition (e.g., THC vs. CBD vs. CBG), dosing, method of consumption (e.g., Topical vs. Inhalation vs. Ingestion) and concurrent tobacco use which may contribute to the observed discrepancies. Although a few small prospective studies and clinical trials have reported encouraging results, more robust human data are urgently needed. Future research should focus on well-designed randomized controlled trials and prospective cohort studies, specifically addressing the following categories: (1) bony healing and fusion, (2) pain management, (3) postoperative complications, and (4) psychological and neuroadaptive outcomes such as tolerance, dependence, and opioid sensitivity.

## 4. Endocannabinoid Physiology and Bony Healing/Fusion

The term “endocannabinoid system” (ECS) was introduced in the 1990s following the discovery of membrane receptors for tetrahydrocannabinol (THC) and their corresponding endogenous ligands [10]. The ECS is now recognized as a complex signaling network composed of cannabinoid receptors, endogenous lipid ligands, and the enzymes responsible for their biosynthesis and degradation [32]. The two primary endocannabinoids are N-arachidonoylethanolamine (anandamide, AEA) and 2-arachidonoylglycerol (2-AG), which act primarily on the CB1 and CB2 cannabinoid receptors (Figure 1). Additional endogenous molecules capable of modulating CB1 and CB2 include: 2-AG ether, N-arachidonoyldopamine, O-arachidonoyl ethanolamine, and oleamide, though they remain less well characterized [32]. Both CB1 and CB2 are G-protein–coupled receptors (GPCRs) expressed at varying levels throughout the body. CB1 is primarily located in the central and peripheral nervous systems, while CB2 is more abundant in the skeletal and immune systems [33].

Physiologically, the ECS plays a critical role in modulating both short- and long-term synaptic plasticity in the central nervous system by regulating excitatory and inhibitory neurotransmission [34,35]. These mechanisms contribute to the modulation of cognition and emotional processing within the cortex, hippocampus, and amygdala [33]. CB1 receptor activation is implicated in pain modulation, cardiovascular regulation, gastrointestinal motility, respiratory control, and the secretion of hypothalamic hormones and peptides, with further modulation by steroid hormone interactions [36,37]. While CB2 was historically less studied, recent research has begun to elucidate its role in immune signaling and inflammation, chronic pain, and skeletal homeostasis [33,38,39,40].

Endocannabinoid concentrations are notably high in both bone and brain tissue, suggesting local production and regulatory function in these environments [41]. Osteoblasts, osteoclasts, and chondrocytes express both CB1 and CB2, underscoring the ECS’s involvement in skeletal physiology [42]. In bone tissue, both CB1 and CB2 receptors are present on osteoblasts, osteoclasts, and other skeletal cells (Figure 1), alongside the enzymatic machinery that regulates endocannabinoid levels. The evolutionary conservation of the ECS across a wide range of mammalian species—and even in more distantly related organisms such as cnidarians—further supports its fundamental biological significance beyond psychotropic effects [43]. This conservation highlights the potential of targeting the ECS in therapeutic applications across diverse tissues.

Importantly, both endogenous and exogenous cannabinoids interact with the ECS in a context-dependent manner. Ligands derived from endogenous sources (endocannabinoids), plants (phytocannabinoids), or synthetic compounds exhibit varying receptor affinities and cell-type-specific selectivity, producing distinct physiological effects. Within the skeletal system, these receptor-ligand interactions are thought to influence bone remodeling, repair, and overall musculoskeletal health [42,44]. Therefore, a deeper mechanistic understanding of ECS signaling and its modulation by plant-derived and synthetic cannabinoids is necessary to fully evaluate their therapeutic potential in bone healing and regeneration. Endogenous cannabinoids such as anandamide (AEA) and 2-arachidonoylglycerol (2-AG) are synthesized on demand by cells (rather than stored in vesicles). In bone tissue, both osteoblasts and osteoclasts can produce AEA and 2-AG locally. Notably, endocannabinoid levels are dynamic and can increase in response to physiological stimuli–for instance, neither AEA nor 2-AG is detectable in normal joint fluid, but both appear in high concentrations in synovial fluid after joint injury, indicating that these signaling lipids are released on-site during trauma. This activity suggests a role for the endocannabinoid system in the acute injury response and healing process. Such findings also parallel age-related changes observed in the ECS; for example, the osteogenic growth peptide (an endogenous CB2 ligand that helps maintain bone formation) declines significantly with age, potentially reducing endocannabinoid “tone” in older individuals [45].

## 5. CB1 Receptor

Cannabinoids exert their effects on bone primarily through the CB1 and CB2 receptors, though other receptors such as GPR55, GPR119, TRPV1, and TRPV4 may also contribute [32,42]. Traditionally, CB1 is referred to as the neuronal cannabis receptor and CB2 is the non-neuronal receptor- although both regulate bone metabolism in different ways. Knock-out, In Vivo and In vitro animal models have demonstrated CB1 as a key component to bone metabolism. Both CB1 antagonists and mouse KO models have been shown to increase bone mass in young animals but exacerbate osteoporotic bone loss in older ones, suggesting CB1 impairs bone maturation during youth while providing protective effects later in life [32,44,46,47,48] (Table 1, Table 2 and Table 3). In fact, CB1 levels in osteoblasts increase with age [46]. CB1 activation has also been associated with enhanced bone formation following trauma to bones such as the calvarium and femur [47]. Tam et al., demonstrated one possible mechanism in which CB1 regulates bone metabolism in which they report CB1 receptors are present in the sympathetic terminals of bone [48]. The mechanism proposed is that Norepinephrine (NE) from sympathetic terminals tonically suppress bone formation via B2 adrenergic receptors but through activation of 2-AG and blockage of NE release, alleviated suppression of bone formation (Figure 2).

Cannabinoids exert their effects on bone primarily through the CB1 and CB2 receptors, though other receptors such as GPR55, GPR119, TRPV1, and TRPV4 may also contribute [32,42]. Traditionally, CB1 is referred to as the neuronal cannabis receptor and CB2 is the non-neuronal receptor- although both regulate bone metabolism in different ways. Knock-out, In Vivo and In vitro animal models have demonstrated CB1 as a key component to bone metabolism. Both CB1 antagonists and mouse KO models have been shown to increase bone mass in young animals but exacerbate osteoporotic bone loss in older ones, suggesting CB1 impairs bone maturation during youth while providing protective effects later in life [32,44,46,47,48] (Table 1, Table 2 and Table 3). In fact, CB1 levels in osteoblasts increase with age [46]. CB1 activation has also been associated with enhanced bone formation following trauma to bones such as the calvarium and femur [47]. Tam et al., demonstrated one possible mechanism in which CB1 regulates bone metabolism in which they report CB1 receptors are present in the sympathetic terminals of bone [48]. The mechanism proposed is that Norepinephrine (NE) from sympathetic terminals tonically suppress bone formation via B2 adrenergic receptors but through activation of 2-AG and blockage of NE release, alleviated suppression of bone formation (Figure 2).

Cannabinoids exert their effects on bone primarily through the CB1 and CB2 receptors, though other receptors such as GPR55, GPR119, TRPV1, and TRPV4 may also contribute [32,42]. Traditionally, CB1 is referred to as the neuronal cannabis receptor and CB2 is the non-neuronal receptor- although both regulate bone metabolism in different ways. Knock-out, In Vivo and In vitro animal models have demonstrated CB1 as a key component to bone metabolism. Both CB1 antagonists and mouse KO models have been shown to increase bone mass in young animals but exacerbate osteoporotic bone loss in older ones, suggesting CB1 impairs bone maturation during youth while providing protective effects later in life [32,44,46,47,48] (Table 1, Table 2 and Table 3). In fact, CB1 levels in osteoblasts increase with age [46]. CB1 activation has also been associated with enhanced bone formation following trauma to bones such as the calvarium and femur [47]. Tam et al., demonstrated one possible mechanism in which CB1 regulates bone metabolism in which they report CB1 receptors are present in the sympathetic terminals of bone [48]. The mechanism proposed is that Norepinephrine (NE) from sympathetic terminals tonically suppress bone formation via B2 adrenergic receptors but through activation of 2-AG and blockage of NE release, alleviated suppression of bone formation (Figure 2).

CB1 and CB2 immunostaining in hypertrophic chondrocytes of the epiphyseal growth cartilage further link CB1 to longitudinal growth and potentially to fracture repair. In vitro demonstrates that CB1 receptor agonists show increased osteogenic differentiation [49,50]. Yan et al., demonstrate mechanistic pathways in which CB1 activation can activate JNK, p38 MAPK and inhibit Erk ½ pathways to rescue mitochondrial metabolism and improve bone regeneration [50]. Aligned with these findings, Gowran et al. show that an antagonist (THC) at CB1 induced apoptosis in bone marrow stem cells, while upregulation of CB1 showed improvements in survival during times of stress [51]. Of note, the CB1 receptor has a myriad of other effects physiologically, with activation leading to weight gain and metabolic syndrome, an important consideration in patients who are either underweight or overweight [50]. Although CB1 activation has demonstrated positive results in prevention of osteoporosis, the exact mechanism remains underexplored and an important area of investigation moving forward.

**Table 1 biomedicines-13-01891-t001:** Murine Knock-Out Models.

Genotype	Phenotype/Effect	Murine Strain	Sex	Age	Citations
CB1 Related Knock-Outs					
CB1−/−	Elevated trabecular & cortical bone mass (young adults)	C57BL/6J	M/F	8–11 weeks	Idris 2005 [52]
CB1−/−	Age-related osteoporosis with low bone formation and fatty marrow	C57BL/6J	M/F	12 months	Idris 2009 [46]
CB1−/−	Suppressed bone formation via heightened sympathetic tone	C57BL/6J	M/F	3 months	Tam 2008 [47]
CB1−/−	Elongated femur & tibia from growth-plate expansion	C57BL/6J	M/F	6 weeks	Wasserman 2015 [53]
CB2 Related Knock-Outs					
CB2−/−	Accelerated trabecular bone loss	C57BL/6J	M/F	8–11 weeks or 51 weeks	Ofek 2006 [54]
CB2−/−	Accelerated Cortical Expansion	C57BL/6J	M/F	8–11 weeks or 51 weeks	Ofek 2006 [54]
CB2−/−	Increased activity of Trabecular Osteoblasts	C57BL/6J	M/F	8–11 weeks or 51 weeks	Ofek 2006 [54]
CB2−/−	Increased Activity of Osteoclasts	C57BL/6J	M/F	8–11 weeks or 51 weeks	Ofek 2006 [54]
CB2−/−	Decreased Diaphyseal Osteoblast precursors	C57BL/6J	M/F	8–11 weeks or 51 weeks	Ofek 2006 [54]
CB2−/−	Increased Cortical Bone	C57BL/6	M/F	3 months	Khalid 2015 [55]
CB2−/−	Reduced capacity to form bone & no protective effects from CB2 agonists	C57BL/6	F	8 weeks	Sophocleous 2011 [56]
CB2−/−	Low Bone Turnover & High Trabecular Bone Mass–In young females	CD1	M/F	12 Months	Sophocleous 2014 [57]
P62 KO	Osteoanabolic Effect	Pg62 Mice	M/F	3 months	Keller 2022 [44]
P62 KO	Increased number of Osteoblasts and Osteoclasts	Pg62 Mice	M/F	3 months	Keller 2022 [44]
CB2−/−	Young female mice had higher trabecular bone mass but lost more with age.	CD1 Mice	M/F	3 months	Sophocleous 2014 [57]
CB2−/−	Similar bone volume decrease compared to wild-type & bone mass reduced with age.	C57BL/6 Mice	M/F	3 months	Sophocleous 2014 [57]

Summary of murine knockout models and their bone-related phenotypes. Abbreviations: M/F = male/female; KO = knockout.

**Table 2 biomedicines-13-01891-t002:** In Vivo Animal Studies.

Study Design	Strain	Sex	Age	Receptor	Ligand	Agonist or Antagonist	Phenotype/Effect	Citation
CB1 Related In Vivo Models								
Ovariectomy-induced bone loss (global KO ± rimonabant)	C57BL/6J	F	Adult	CB1	Rimonabant	Antagonist	Bone mass preserved; osteoclast suppression	Idris 2005 [52]
Mild TBI remote bone-gain model	C57BL/6J	M	12 weeks	CB1	2-AG	Agonist	CB1-dependent bone formation; absent in KO	Tam 2008 [47]
Glucocorticoid osteoporosis (methyl-pred ± rimonabant)	Sprague-Dawley	M	3–4 mo. & 12–14 mo.	CB1	Rimonabant	Antagonist	Antagonist prevents loss in young; worsens loss in old	Samir 2014 [58]
Calvarial TBI bone-gain model ± rimonabant	ICR & C57BL/6J	M	6 & 12 weeks	CB1	Rimonabant	Antagonist	Skull bone gain CB1-dependent; Antagonist blocks	Eger 2019 [59]
Diabetic bone loss, renal-tubule CB1 KO	C57BL/6J	M	Adult	CB1	CB1 antag	Antagonist	KO/antagonist prevent loss via increased EPO and formation	Baraghithy 2021 [60]
CB2 Related In Vivo Models								
Ovariectomy-Induced Bone Loss	C57BL/J	M/F	3 vs. 6 months	CB2	OGP	Agonist & Allosteric Modulator	Age-Related Decline in Bone Mass	Rapheal-Mizrahi 2022 [45]
Glucocorticoid induced Osteonecrosis	Sprague Dawley	M	10 weeks	CB2	JWH133	Agonist	Alleviation of Glucocorticoid induced Osteonecrosis	Sun 2021 [61]
Glucocorticoid induced Osteonecrosis	Sprague Dawley	M	10 weeks	CB2	JWH133	Agonist	Osteogenic protection via GSK/3B/B-Catenin signaling pathway	Sun 2021 [61]
Glucocorticoid induced Osteonecrosis	Sprague Dawley Rat	M	10 weeks	CB2	JWH133	Agonist	Stimulation of Angiogenesis	Sun 2021 [61]
Ovariectomy-Induced Bone Loss	C57BL/6J Mice	M/F	10 weeks	CB2	HU-308	Agonist	Protective of Bone Loss	Smoum 2015 [62]
Ovariectomy-Induced Bone Loss	C57BL/6J Mice	M/F	10 weeks	CB2	HU- 433	Agonist	Protective of Bone Loss (3–4 order of magnitude more potent)	Smoum 2015 [62]
Ovariectomy-Induced Bone Loss	Mice	F	10 weeks	CB2	2-AG	Agonist	Up-regulated Notch 1 Expression and promoted differentiation of hbMSCSs	Tian 2021 [63]
Ovariectomy-Induced Bone Loss	C3H Mice	F	8 or 51 weeks	CB2	HU-308	Agonist	Attenuates Bone loss, stimulated cortical thickness, suppresses osteoclast number and stimulates endocortical bone formation. Suppressed Osteoclast genesis. By reducing the availability of RANKL.	Ofek 2006 [54]

Summary of in vivo animal studies examining effects of cannabinoid receptor modulation on bone outcomes. Abbreviations: TBI = traumatic brain injury; KO = knockout; EPO = erythropoietin; mo. = months.

**Table 3 biomedicines-13-01891-t003:** In Vitro Studies.

Cell Line	Cell-Type	Receptor	Ligand	Agonist or Antagonist	Effect	Citation
PDLSCs	MSCs (periodontal)	CB1	R(+)-Methanandamide	Agonist	Increased osteogenic differentiation (TNF/IFN-resistant)	Yan 2019 [49]
hBMSCs	MSCs (bone marrow)	CB1	CB1 over-expression	Genetic OE	Increased mineralization; rescued TNF/IFN inhibition	Yan 2022 [50]
Mouse BMSCs	MSCs (bone marrow)	CB1	SR141716A (Rimonabant)	Antagonist	Induced apoptosis; decreased mineralized matrix	Gowran 2013 [51]
CD14⁺-derived OCs	Osteoclasts	CB1	Anandamide (AEA)	Agonist	Increased osteoclast formation + resorption	
BMSCs	MSCs (bone marrow)	CB2	JWH133	Agonist	Osteogenic protection Via GSK/3B/B-Catenin signaling pathway	Sun 2021 [61]
BMSCs	MSCs (bone marrow)	CB2	JWH133	Agonist	Stimulation of Angiogenesis Via endothelial cell migration & VEGF	Sun 2021 [61]
C57BL/6 Mice	BMSCs	CB2	CBD	Agonist	Osteogenic Differentiation Via p38 MAPK signaling pathway	Li 2022 [64]
Human Bone Marrow	Osteoblasts	CB2	JWH-133	Agonist	Improves Osteogenesis	Rossi 2015 [65]
Mouse Calvarial cells	Osteoblasts	CB2	HU-308/HU-433	Agonist	Protection of Bone Loss	Smoum 2015 [62]
MC3T3 E1	Osteoblasts	CB2	HU-308	Agonist	Osteoblast Proliferation Via Erk1/2 phosphorylation and Map-kapk2 protein synthesis in G protein cyclin D1 mitogenic axis	Ofek 2011 [66]
BMSCs & RAW264.7 cell line	Osteoblasts & Osteoclasts	CB2	Desmethoxyyangoni, Flavokawain A, Echinatin, Mangiferin, 11-keto-B-boswelic acid	Agonist	Promote Osteogeneis & Inhibit Osteoclast Differentiation	Hu 2022 [67]
Human Osteoblasts	Osteoblasts	CB2	OGP	Agonist	Proliferation of Osteoblasts	Raphael-Mizrahi 2022 [45]
Mouse BDMS	BDMS	CB2	OGP	Agonist	Attenuation of Osteoclast Differentiation	Raphael-Mizrahi 2022 [45]
Mouse BDMS	BDMS	CB2	OGP	Agonist	Anti-Inflammatory Activity in Macrophages	Raphael-Mizrahi 2022 [45]
Mouse BMDS	BDMS & Osteoblasts	CB2	HU-308	Agonist	Direct stimulation of stromal cells/osteoblasts and inhibition of monocytes/osteoclasts by direct inhibition of RANKL expression	Ofek 2006 [54]
Human Bone Marrow	Osteoblasts	TRPV1	Resiniferatoxin (RTX)	Agonist	Improves Osteogenesis	Rossi 2015 [68]

Summary of in vitro studies on cannabinoid receptor effects. Abbreviations: MSCs = mesenchymal stem cells; BDMS = bone-derived mesenchymal stromal cells; IFN = interferon; TNF = tumor necrosis factor; OE = over-expression; OGP = osteogenic growth peptide.

## 6. CB2 Receptor

The CB2 receptor is found in bone cells at higher levels than CB1 receptors [51]. In cortical bone, CB2 activation drives endosteal formation by stimulating osteoblast proliferation, differentiation and matrix deposition [69]. Developmentally, CB2 also contributes to bone elongation [70]. One study further demonstrated that CB1 and CB2 are expressed in hypertrophic chondrocytes within the epiphyseal growth cartilage (EGC), which drives vertebrate skeletal growth and could potentially be implicated in bone healing [53]. CB2 receptor activation consistently promotes osteogenesis by enhancing osteoblast activity, reducing osteoclast formation, and accelerating fracture repair; conversely, CB2 knock-out models exhibit osteoporotic, high-turnover bone phenotypes [54]. Ofek et al., demonstrate that CB2 deficient mice have age accelerated trabecular bone loss, while CB2 agonists having a positive effect on maintaining bone mass via osteoblastic and osteoclastic CB2 signaling [54]. Four widely used synthetic CB2-selective agonists—HU-308, HU-433, JWH-133 and JWH-015 enhance osteoblast activity, whereas the synthetic antagonists AM-630 and SR-144528 curb osteoclast differentiation and function [52,56,71,72]. Mixed cannabinoid agonists such as WIN55,212-2 have also demonstrated neutral or mildly beneficial effects on spinal fusion outcomes, suggesting that CB2-biased or CBD-rich cannabinoid regimens may support early bone formation without compromising final fusion integrity [73]. Furthermore, alternative circulating peptides- Osteogenic growth peptide (OGP) shows evidence of ameliorating age-related bone loss by maintaining a skeletal “CB2 tone [73].” Of note, circulating OGP declines significantly with age, indicating endogenous peptide signaling may have important roles in skeletal human health [45].

CB2-related knock-out models demonstrate overall that CB2 is essential for bone regulation and growth, although there are a few key nuances. Ofek et al., demonstrate accelerated trabecular bone loss but accelerated cortical expansion in a CB2 −/− model [54]. Despite some conflicting findings, overall removal of CB2 receptor results in increased activity of osteoclasts and decreased osteoblast precursors. Furthermore, CB 2 −/− in young mice showed increased cortical bone but in older mice showed high trabecular bone [57]. Sophocleous et al., demonstrate that CB2 receptor has gender and age specific effects that also depend on genetic differences between mouse strains [1]. Moreover, P62, an regulatory site, showed osteoanabolic effects and increased numbers of osteoblasts and osteoclasts when the gene was knocked out [44].

Cannabis use is increasingly prevalent among patients with spinal disorders, yet its influence on bone healing and spinal fusion remains complex and incompletely defined. In vitro studies showed similar results, with activation of CB2 resulting in osteogenesis, proliferation of osteoblasts, attenuation of osteoclast differentiation and anti-inflammatory activity of macrophages [45,67]. Moreover, Sun et al. showed osteogenic protection via the GSK/3B/B-Catenin pathways and stimulation of angiogenesis [53,67]. Mechanistically, this is still being elucidated but these studies show GSK/3B/B-Catenin, VEGF, Erk ½, and inhibition of receptor activator of nuclear factor kappa-B ligand (RANKL) to be part of our current understanding [54,61,64]. In a rat posterolateral spinal fusion model, activation of CB2 receptors significantly increased new bone formation at the fusion site, consistent with CB2’s osteogenic mechanism of action. Mechanistically, CB2 signaling promotes Wnt/β-catenin pathways and angiogenesis (via VEGF), while suppressing osteoclastogenesis through RANKL inhibition, which together drive a more robust fusion mass. It is important to highlight the varied effects different ligands have at CB2. OGP showed age-related decline in bone mass. In murine experiments, the synthetic CB2 agonist JWH-133 alleviated glucocorticoid-induced bone loss via activation of β-catenin signaling [61]. HU-308 had similar results in which it had protection of bone loss, with HU-433 have 3–4 orders of magnitude more bone loss protection [62]. This demonstrates that, not only activation of CB2 needs to be accounted for but variations in binding and even other modulatory sites. This makes further research key- as cannabis in public use varies widely in strains, dosages, and composition.

### 6.1. Cannabidiol (CBD)

Cannabidiol (CBD) is a weak antagonist for CB1 and CB2 receptors, but also interacts with TRPV1, TRPV2, GPR55, and dopamine receptors [74]. Numerous studies support CBD’s role in bone homeostasis, fracture healing, and regeneration. One study in a rat spinal cord injury model found that CBD attenuated sub lesional cancellous bone loss, enhanced bone volume and thickness, and improved mechanical properties of bone [70]. Li et al., demonstrated that CBD upregulated mRNA expression of alkaline phosphatase and osteoprotegerin and downregulated expression of NF-kB ligand- showing positive potential for SCI induced bone loss. Another showed that CBD alone significantly enhanced the mechanical strength of fracture callus, although it did not increase bone volume or mineral content [11]. Additional research confirmed that CBD promotes osteogenic differentiation of mesenchymal stem cells (MSC), resulting in improved bone bridging [75].

During early inflammation, CBD was shown to accelerate mineralization of the fibrocartilaginous callus and increase the viability and proliferation of bone and bone marrow cells [76]. These effects led to higher bone volume fraction, increased bone mineral density, and improved mechanical strength of newly formed bone. The results were promising enough to suggest that CBD could potentially replace NSAIDs for post-fracture pain management while simultaneously promoting bone repair [76].Mechanistically, CBD promotes bone regeneration through the upregulation of osteogenic genes such as ALPL, BMP4, and RUNX2 [73]. In osteoporotic animal model populations, CBD has been shown to improve bone morphology, although further research is warranted [77].

Both THC and CBD demonstrate dose-dependent effects on human osteoclast fusion and bone resorption [78]. At lower doses, osteoclast fusion was unaffected while bone resorption increased; at higher doses, both osteoclast fusion and resorption were inhibited. In co-culture systems, both osteoclastic activity and alkaline phosphatase activity of osteoblast lineage cells were suppressed. Cannabinoid receptor CNR2 is more highly expressed than CNR1 in CD14+ monocytes and pre-osteoclasts; differentiation into mature osteoclasts was associated with decreased CNR2 expression. Under co-culture conditions, expression of CNR2,but not CNR1,was detected in both osteoclast and osteoblast nuclei [78].

Overall, relative to other cannabinoids, besides THC, CBD is the most well studied. CBD has a very safe profile, is non-intoxicating and does not reinforce cravings [79]. Two interventional clinical, human studies conducted by Kulpa et al., demonstrated positive effects- such as attenuation of bone loss. Despite some evidence of improved bone turnover, the results were inconclusive [29,30]. The preclinical data overall has shown positive effects in vitro and in vivo but more mechanistic studies and randomized clinical trials need to be done to demonstrate CBD’s overall potential, although currently promising.

### 6.2. Cannabigerol (CBG)

Cannabigerol (CBG) is a major phytocannabinoid found in Cannabis that is gaining interest due to its non-psychotropic effects. It functions as a partial agonist at CB2 receptors, with measurable, though not fully elucidated, activity at CB1 [80]. Although not traditionally studied in the context of bone healing, one study demonstrated that CBG attenuates neuropathy-induced hypersensitivity, particularly in models of neuropathic rather than thermal or persistent pain [81]. In a fracture model, CBG was associated with improved bone volume fraction, bone mineral density, and mechanical strength [76]. Most interesting about this study is that CBG demonstrated healing mechanisms and outcomes comparable to CBD regarding bone healing.

Few studies have addressed cannabinoid pharmacokinetics, though it is known that CBG is the biosynthetic precursor to other cannabinoids. Cannabigerolic acid (CBGA), the native plant form of CBG, has been shown to be absorbed 40 times more efficiently than CBG in dogs, regardless of fed or fasted state [82].These findings underscore the importance of cannabinoid formulation, dosage, and route of administration in determining bioavailability and therapeutic efficacy. Although THC and CBD hold most of the space regarding research of bone healing, these early studies show that CBG can also help in not only bone healing but also limit pain. Our review highlights the importance of further research in all types of cannabinoids and cannabis formulations, as they can potentially provide additive or complimentary effects that could benefit patients undergoing surgery or healing from bone fractures.

### 6.3. Tetrahydrocannabinol (THC)

Tetrahydrocannabinol (THC), the primary psychoactive constituent of cannabis, functions as a partial agonist at both CB1 and CB2 receptors [42]. Its effects on skeletal development and bone healing are multifaceted, with distinct age, sex, and dose-dependent implications. In preclinical studies, THC administration slowed skeletal elongation through CB1 activation in female rats, while males were unaffected. THC exposure was also associated with reduced body weight [53]. These findings are consistent with prior evidence suggesting that cannabis may exert anti-estrogenic effects [83]

Developmental exposure to THC, particularly during pregnancy, has been shown to impair intrauterine, femoral, and vertebral growth in animal offspring [53]. In juvenile animal models, THC inhibits chondrocyte hypertrophy and endochondral ossification, primarily through CB1 receptor signaling [53]. This reflects suppression of growth plate activity and disruption of normal skeletal development. In the context of bone healing, THC exhibits a biphasic response. In vitro, studies demonstrate that lower doses enhance human osteoclast resorption, while higher doses inhibit both osteoclast and osteoblast activity [78]. Chronic THC exposure has also been shown to reduce MSC viability and impair osteogenic differentiation [51]. Additionally, THC may attenuate the bone-promoting effects of cannabidiol (CBD) at later stages of healing, indicating a complex interaction between cannabinoids that depends on timing and dosage [11].

These mechanistic findings are supported by clinical data. Retrospective clinical studies have linked heavy cannabis use with higher pseudarthrosis rates in spine fusion patients. Chronic heavy cannabis use has been associated with reduced bone mineral density, typically in the range of 6 to 10 percent, and a significantly higher rate of pseudarthrosis following spinal fusion surgery [26]. One study reported up to a 3.6-fold increase in revision surgery risk among heavy users [25,26]. Although there is a need for prospective randomized controlled trials to confirm retrospective data and preliminary basic science findings, clinicians should provide careful counseling on the skeletal risks associated with THC use. Both dosage and chronicity may serve as modifiable risk factors that should be considered in a manner like tobacco use when evaluating bone health and surgical outcomes. In contrast, non-psychoactive cannabinoids such as CBD or CB2-selective agonists may offer opioid-sparing analgesic benefits without impairing fusion outcomes [73].

Future research should prioritize prospective trials that clearly define THC/CBD ratios, optimal dosage thresholds, and cannabinoid formulations, and that incorporate pharmacokinetic monitoring to correlate dosing with drug levels and healing outcomes. These studies must also account for confounding factors. For example, researchers should use appropriate statistical approaches (multivariable regression, propensity score matching, etc.) to control for co-variables like tobacco use or concurrent medications, thereby isolating the effects of cannabis itself. A targeted and personalized approach will be essential to safely integrate cannabinoid science into perioperative care while maximizing therapeutic potential and minimizing skeletal risk.

Future studies must address three key areas to translate preclinical insights into clinical practice. First, researchers need to identify which specific cannabinoids and at what doses most reliably accelerate bone healing. Early evidence indicates that CBD consistently promotes osteoblast activity, whereas chronic, high-dose THC may impair fusion; compounds such as CBG also show promise but require head-to-head comparison in well-controlled animal models. Second, it is essential to clarify how confounding factors modify these effects. Tobacco co-use (for example, cannabis mixed with nicotine), and concurrent opioid consumption could alter the bone-repair process, and different delivery methods (smoking, edibles, topicals) may influence pain control, inflammation, and local tissue healing. Third, developing simple, noninvasive biomarkers such as serum collagen crosslinking enzymes or bone turnover markers will allow clinicians to track fracture or fusion progress in real time. By conducting rigorous, multimodal studies that include healthy adults as well as high-risk groups (osteoporotic, elderly, or postmenopausal patients), the field can establish evidence-based guidelines for leveraging cannabinoids in spinal fusion and fracture care.

## 7. Conclusions

In summary, preclinical data indicate that cannabinoids influence bone healing through the endocannabinoid system, with an overall trend toward positive effects. Given the limitations of available studies (often small, retrospective, and subject to confounding), these data should be regarded as preliminary. Prospective research will be needed to confirm whether the observed associations are truly causal.

Despite the breadth of data summarized, significant uncertainties and conflicting evidence remain. Preclinical models and clinical studies do not always agree; for instance, animal experiments often show cannabinoid-driven bone formation, whereas human observational studies sometimes link cannabis use to impaired healing. These discrepancies highlight key controversies in the field and underscore the influence of context (dose, duration, and individual patient factors) on outcomes. In our view, chronic heavy THC exposure emerges as a modifiable risk factor for nonunion, like tobacco use, even though short-term or low-dose cannabinoid exposure might be benign or even beneficial. We advise interpreting the current evidence with caution: findings are intriguing but not yet definitive, and mechanistic complexities (e.g., biphasic dose responses, poly-cannabinoid interactions) mean that further research is needed before drawing firm clinical conclusions.

Cannabidiol (CBD) consistently enhances osteoblast activity and callus formation, whereas Δ9-tetrahydrocannabinol (THC) yields mixed results, promoting early ossification at low doses but impairing fusion and reducing bone mineral density when used chronically or at high concentrations. Emerging compounds such as cannabigerol (CBG) and other CB1/CB2 receptor modulators also show promise but remain insufficiently studied in both in vitro and in vivo settings. These findings support the hypothesis that a multi-cannabinoid signaling axis could synergistically enhance spinal fusion and fracture repair, though the specific mechanisms and long-term consequences of such interactions are not yet fully understood.

As cannabis use becomes more widespread, it is essential to clarify how real-world variables impact these effects. Confounding factors such as tobacco co-use in the form of blunts and concurrent opioid consumption may modify skeletal repair and fusion success. Different routes of administration (smoking, edibles, topicals) could alter pain control, anxiety management, and local tissue healing in spine surgery patients, and the anxiolytic or analgesic properties of specific cannabinoids must be evaluated within rigorous clinical frameworks. Clinicians should approach perioperative cannabis use proactively. While formal guidelines await higher-quality evidence, current data suggest it is wise to treat heavy cannabis use as a potential risk factor for impaired fusion much like smoking. We recommend that surgeons screen for cannabis use in their patients and engage in frank discussions: patients who heavily use THC-rich cannabis might be advised to taper or abstain in the perioperative period to maximize healing potential. Conversely, patients using predominantly CBD based products for analgesia (with little or no THC) may not face the same bone healing risk, though this too should be monitored. Key patient factors such as concomitant tobacco use, baseline bone density, and possibly sex or genetic differences in cannabinoid receptors should be considered when interpreting the impact of cannabis on an individual’s healing. In practice, a reasonable approach is to individualize counseling and follow-up: for a young patient occasionally using cannabis, evidence does not currently mandate a strict prohibition, but for an older or high-risk patient with daily marijuana use, a more cautious plan (including advising cessation and scheduling closer post-op radiographic checks) is warranted.

From a surgical perspective, patients should be counseled on cannabis use just as they are on smoking. In the absence of definitive guidelines, a prudent approach is to advise patients to limit or refrain from cannabis (especially high THC products) in the weeks before and after spinal fusion surgery. Surgeons are encouraged to document cannabis use in the preoperative work up and consider patients’ use patterns (e.g., casual vs. daily heavy use) when planning postoperative follow-up. Heavy cannabis users, particularly those who also smoke tobacco, may merit closer radiographic monitoring for fusion progress, given the higher pseudarthrosis rates observed. Conversely, the use of CBD dominant products could be explored for pain management, as these are less likely to impair bone healing (though this should be weighed on a case-by-case basis). Overall, we suggest treating perioperative cannabis use as a modifiable risk factor: through patient education, potentially short-term cessation programs, and personalized risk benefit discussions. Future work should also delineate how individual cannabinoids and strains (CBD, CBG, THC) diverge in their biological effects, identify reliable biomarkers like collagen crosslinking enzymes or bone turnover markers to monitor healing, and establish dose response relationships across diverse populations including healthy adults, women, young versus elderly cohorts, osteoporotic patients, and those undergoing orthopedic and neurosurgical procedures. Addressing these questions methodically will facilitate the translation of preclinical findings into robust, evidence-based clinical guidelines that enhance bone healing and surgical outcomes. It should also be noted that access to medical cannabis is uneven patients in regions where cannabis remains illegal or unavailable do not have the option to legally use these therapies for pain control, which raises equity concerns in postoperative care.

## Figures and Tables

**Figure 1 biomedicines-13-01891-f001:**
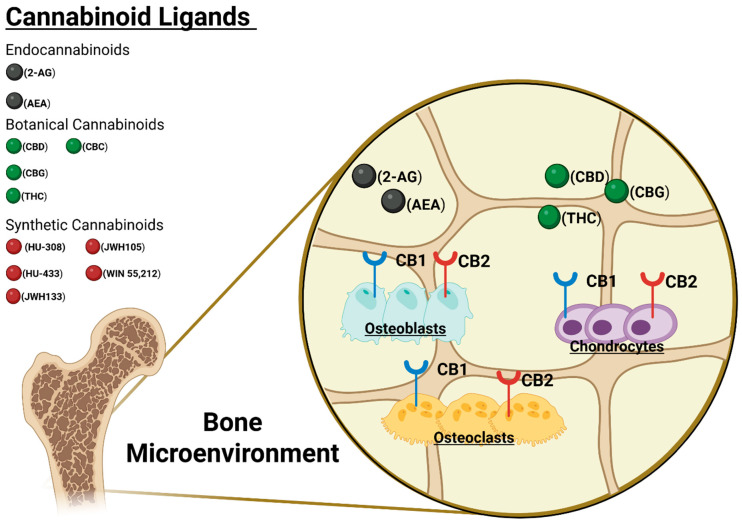
A schematic of the endocannabinoid system and receptor distribution in bone tissue. The figure shows CB1 (red) and CB2 (blue) receptors on osteoblasts and osteoclasts, along with their endogenous ligands (anandamide and 2-AG) and the enzymes that synthesize and degrade these ligands. This schematic highlights how ECS components are present in bone cells and may regulate bone remodeling.

**Figure 2 biomedicines-13-01891-f002:**
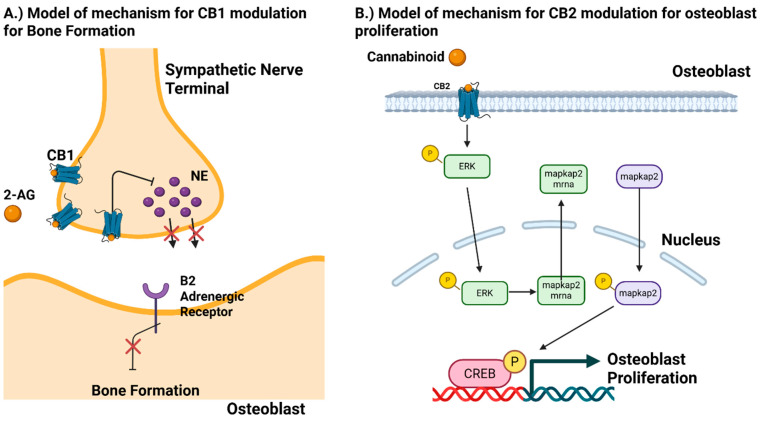
Effects of CB1 and CB2 receptor modulation on osteoblasts, osteoclasts, and mesenchymal stem cell differentiation. CB1 and CB2 signaling influence bone remodeling via distinct pathways: CB2 activation (**B**) directly stimulates osteoblast differentiation and activity (promoting new bone formation) while inhibiting osteoclast development and function (reducing bone resorption), in part by down-regulating RANKL and up-regulating pro-osteogenic factors (e.g., Wnt/β-catenin and VEGF signaling). CB1 activation (**A**) acts largely through the nervous system: activation of CB1 on sympathetic nerve terminals can indirectly enhance osteogenesis by reducing adrenergic (norepinephrine-mediated) inhibition of osteoblasts; however, excessive CB1 stimulation (for example, by high-dose THC) can suppress both osteoblast and osteoclast activity, reflecting a dose-dependent biphasic effect on bone remodeling.

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
