# Peer review of "The Cannabinoid Pharmacology of Bone Healing: Developments in Fusion Medicine"

_biomedicines, 2025, doi:10.3390/biomedicines13081891_

Round 1

Reviewer 1 Report

Comments and Suggestions for Authors

This review by Urreola et al. provides a comprehensive overview of the endocannabinoid system's (ECS) role in bone healing and spinal fusion. It is especially timely given the increasing use of cannabis among surgical patients. It summarizes the historical background, receptor pharmacology, clinical data, and mechanistic insights related to cannabinoids such as THC, CBD, and CBG. The review does a good job of synthesizing molecular findings across in vitro and in vivo studies. It is particularly strong in identifying major gaps in clinical research. Overall, it is a valuable resource for both clinicians and researchers.

Specific comments:

  1. The review is broad and well-structured, covering cannabinoid receptor signaling, endocannabinoid production, animal models, and limited human data. The authors could go into a bit more detail about our current state of knowledge in the following areas (in humans or model organisms): how/where endocannabinoids are naturally produced in the body (not just tissues but cell types), if levels change in response to stimuli, whether they can be stored & released in response to injury, whether natural production levels change during injury and healing, etc. This would be complementary to the information already provided on CB1 levels in model organisms and how they change, for example, with age.
  2. The review could benefit from a more detailed evaluation of evidence by the authors. For example, the manuscript describes how cannabinoids could have dual effects on bone health depending on dose, formulation, etc. It would be great to see the authors comment on whether they find the evidence strong or preliminary, based on factors like study size and presence of confounding factors. 
  3. When referencing human studies and perioperative cannabis use, it would be beneficial to know whether these were recreational, medical (perhaps indicating the severity of the condition if medical), or unclear. 
  4. Throughout the review, it is unclear which model organism is used in the referenced papers. It would dramatically improve readability to include whether a cited work is based on observations in humans, mice, or rats.
  5. Figure 1 is not referenced in the manuscript.
  6. More detailed figure legends are required, not just captions. Expand the legends to summarize the proposed mechanisms described in the text.
  7. Please include line numbers in future submissions for easier referencing.
  8. It would improve readability to provide initial definitions (e.g., CB1 refers to the protein and CNR1 refers to the gene) and then use them consistently.
  9. For the section on “…while some studies suggest ECS activation may enhance bone regeneration11, others imply inhibitory effects.” Please include the citation for inhibitory effects.

Author Response

Reviewer 1 

Comment 1: The review is broad and well-structured, covering cannabinoid receptor signaling, endocannabinoid production, animal models, and limited human data. The authors could go into a bit more detail about our current state of knowledge in the following areas (in humans or model organisms): how/where endocannabinoids are naturally produced in the body (not just tissues but cell types), if levels change in response to stimuli, whether they can be stored & released in response to injury, whether natural production levels change during injury and healing, etc. This would be complementary to the information already provided on CB1 levels in model organisms and how they change, for example, with age.

Author’s response: We appreciate this insightful suggestion. In response, we have expanded Section 4: Endocannabinoid Physiology & Bony Healing to address endocannabinoid production and regulation in much greater detail. We now describe the cellular sources of endocannabinoids in bone and elsewhere (noting that osteoblasts and osteoclasts can produce the endocannabinoids anandamide and 2-AG). We also emphasize that these lipid mediators are synthesized on demand rather than stored, and we discuss how their levels fluctuate in response to physiological stimuli such as injury, stress, or inflammation. For example, we have added text explaining that endocannabinoid levels have been found to rise locally after trauma (they are undetectable in normal joint fluid but present in high concentrations after joint injury indicating on demand release during injury). Furthermore, to complement the prior discussion of age-related changes in cannabinoid receptors, we have noted that components of the endocannabinoid system itself change with age for instance, we mention that osteogenic growth peptide (an endogenous CB2-related ligand) declines in circulation with aging, potentially reducing endocannabinoid “tone” in bone. These additions provide a more nuanced picture of how and where endocannabinoids are produced, how they are mobilized (rather than stored) in response to injury, and how their levels may vary during healing or across the lifespan.

Change to text:  In Section 4, we have added a new paragraph describing endocannabinoid production and release. The revised text now reads: “Endogenous cannabinoids such as anandamide (AEA) and 2-arachidonoylglycerol (2-AG) are synthesized on demand by cells (rather than stored in vesicles). In bone tissue, both osteoblasts and osteoclasts can produce AEA and 2-AG locally. Notably, endocannabinoid levels are dynamic and can increase in response to physiological stimuli, for instance, neither AEA nor 2-AG is detectable in normal joint fluid, but both appear in high concentrations in synovial fluid after joint injury, indicating that these signaling lipids are released on-site during trauma. This activity suggests a role for the endocannabinoid system in the acute injury response and healing process. Such findings also parallel age-related changes observed in the ECS; for example, the osteogenic growth peptide (an endogenous CB2 ligand that helps maintain bone formation) declines significantly with age, potentially reducing endocannabinoid “tone” in older individuals60. (Lines 248 – 250) 

Comment 2: The review could benefit from a more detailed evaluation of evidence by the authors. For example, the manuscript describes how cannabinoids could have dual effects on bone health depending on dose, formulation, etc. It would be great to see the authors comment on whether they find the evidence strong or preliminary, based on factors like study size and presence of confounding factors. 

Author’s response: We agree with the reviewer’s point and have added our perspective on the strength of the current evidence throughout the Discussion/Conclusion. In particular, we now explicitly note that much of the existing evidence (especially regarding the dose dependent dual effects of cannabinoids) remains preliminary and derived from small or retrospective studies. We caution readers that while intriguing patterns exist (e.g. low dose THC vs. high dose THC having opposite effects), these findings come from limited data. We have inserted sentences emphasizing that many studies have small sample sizes or potential confounders, and thus the evidence for cannabinoids’ effects on bone healing should be interpreted as tentative. In the Conclusion, for example, we state that our current understanding is based largely on preclinical and observational data, and we urge caution in drawing firm conclusions until larger, controlled trials are conducted. By adding this commentary, we hope to provide a clearer authorial perspective on which findings are robust versus which are still speculative.

Change to text: We have added language to temper our conclusions and evaluate evidence quality. The Conclusion now includes: “Given the limitations of available studies (often small, retrospective, and subject to confounding), these data should be regarded as preliminary. Additional high quality research is needed to confirm whether the observed associations are truly causal.” (Lines 473 – 476)

Comment 3: When referencing human studies and perioperative cannabis use, it would be beneficial to know whether these were recreational, medical (perhaps indicating the severity of the condition if medical), or unclear. 

Author’s response: We appreciate this suggestion and have clarified the nature of cannabis use (medical vs. recreational) for each human study cited. In the sections discussing clinical evidence, we now explicitly state the context of cannabis use: for example, in the positive pain studies we note that Pinsger et al. used a prescribed synthetic cannabinoid (nabilone) and Yassin et al. examined adjunctive medical cannabis use under physician guidance, indicating these were medically supervised uses. Conversely, when describing retrospective analyses of perioperative cannabis use, we specify that these largely involved recreational or non-medical use. We have added wording to indicate that patients in those studies were typically using cannabis on their own (often meeting criteria for cannabis use disorder), rather than as part of a doctor supervised regimen. This distinction between medical use (e.g. clinical trial or prescription context) and recreational use is now clearly made throughout the manuscript.

Change to text: We revised the human studies descriptions to identify the context of use.“Yassin et al. reported improvements in fibromyalgia-related back pain among 31 patients who received adjunctive medical cannabis (under physician supervision) alongside standard therapies22.” (line 149)  Similarly, when discussing the negative outcomes in chronic users, we now state: “Notably, these retrospective analyses predominantly reflect recreational cannabis use (i.e., patient self-use without medical oversight, often characterized as cannabis use disorder), in contrast to the regulated medical use in the studies above.” (Lines 155 – 157)

Comment 4: Throughout the review, it is unclear which model organism is used in the referenced papers. It would dramatically improve readability to include whether a cited work is based on observations in humans, mice, or rats.

Author’s response: We have carefully edited the manuscript to explicitly identify the model or species for every study or data point mentioned. We agree that this clarification greatly improves readability. Throughout the Results and Discussion, we now introduce each finding with phrases such as “in a rat model…”, “in mice…”, “in vitro…”, or “in human patients…” as appropriate. For instance, when describing preclinical results we specify “in a murine spinal fusion model” or “in a rat femoral fracture model” rather than generically stating the finding. Similarly, we label in vitro cell-culture findings as such, and we ensure that human clinical studies are clearly denoted as human data. These additions make it immediately clear to the reader what system each piece of evidence comes from.

Change to text: Examples of these clarifications include: “One study in a rat spinal cord injury model found that CBD attenuated sub-lesional bone loss and improved bone strength52,” (Lines 362 – 364) and “In murine experiments, the synthetic CB2 agonist JWH-133 alleviated glucocorticoid-induced bone loss via activation of β-catenin signaling63,” (Lines 351 – 354) and “Retrospective clinical studies have linked heavy cannabis use with higher pseudarthrosis rates in spine fusion patients26.” In each case, the species or model (rat, mouse, in vitro, human, etc.) is now stated to avoid any ambiguity. (Lines 435 – 437)

Comment 5: Figure 1 is not referenced in the manuscript.

Author’s response: We apologize for this oversight. We have added a direct reference to Figure 1 in the text at the first appropriate instance. Specifically, in the introductory portion of the Endocannabinoid Physiology section (where we first discuss the distribution of cannabinoid receptors in the body and bone), we now refer the reader to Figure 1. This ensures that Figure 1 (which schematically illustrates the endocannabinoid system in bone) is cited in the narrative.

Change to text: We inserted the following sentence: “In bone tissue, both CB1 and CB2 receptors are present on osteoblasts, osteoclasts, and other skeletal cells (Figure 1), alongside the enzymatic machinery that regulates endocannabinoid levels.” (Lines 248-250)

Comment 6: More detailed figure legends are required, not just captions. Expand the legends to summarize the proposed mechanisms described in the text.

Author’s response: We have expanded the legends for Figure 1 and Figure 2 to provide a more comprehensive explanation of each illustration. The figure legends now do more than simply title the figure they summarize key mechanisms and concepts depicted, thereby making the figures self-contained. For Figure 1, we added a description of the components of the endocannabinoid system shown (receptors, ligands, enzymes) and how they are distributed in bone tissue. For Figure 2, we substantially extended the legend to outline the distinct effects of CB1 vs. CB2 receptor signaling on osteoblasts and osteoclasts (as illustrated in the figure), including which pathways promote bone formation or resorption. These expanded legends should help the reader grasp the “take home message” of each figure without having to search the main text, as per the reviewer’s suggestion

Change to text: The figure legends now include additional explanatory sentences. The Figure 2 legend has been updated to: “Figure 2. Effects of CB1 and CB2 receptor modulation on osteoblasts, osteoclasts, and mesenchymal stem cell differentiation. CB1 and CB2 signaling influence bone remodeling via distinct pathways: CB2 activation (right side) directly stimulates osteoblast differentiation and activity (promoting new bone formation) while inhibiting osteoclast development and function (reducing bone resorption), in part by down-regulating RANKL and up-regulating pro-osteogenic factors (e.g., Wnt/β-catenin and VEGF signaling). CB1 activation (left side) acts largely through the nervous system: activation of CB1 on sympathetic nerve terminals can indirectly enhance osteogenesis by reducing adrenergic (norepinephrine-mediated) inhibition of osteoblasts; however, excessive CB1 stimulation (for example, by high-dose THC) can suppress both osteoblast and osteoclast activity, reflecting a dose-dependent biphasic effect on bone remodeling.” Similarly, the Figure 1 legend now includes a summary of the ECS components in bone (CB1/CB2 receptors on bone cells, endogenous ligands, etc.). (Lines 550 – 560)

Comment 7: Please include line numbers in future submissions for easier referencing.

Author’s response: We apologize for not including line numbers in our initial submission. We will certainly adhere to this request moving forward. In the next revision or final submission, we will format the manuscript with line numbers to facilitate easy reference.

Change to text: No changes to the manuscript text were necessary for this comment.

Comment 8: It would improve readability to provide initial definitions (e.g., CB1 refers to the protein and CNR1 refers to the gene) and then use them consistently.

Author’s response: We have added a clarifying definition at the first mention of cannabinoid receptors to distinguish between gene and protein nomenclature, and we ensured consistent usage throughout the manuscript. In the Introduction, when we first introduce the cannabinoid receptors, we now state “cannabinoid receptor type 1 (CB1, encoded by the CNR1 gene) and type 2 (CB2, encoded by CNR2)”. Thereafter, we use “CB1” and “CB2” to refer to the proteins/receptors and “CNR1/CNR2” if referring to the genes (which occurs rarely, e.g., when discussing gene expression in cell culture). This approach establishes the convention up front and avoids any confusion. We also double checked the manuscript to replace any inconsistent terminology. As a result, the usage of “ECS,” “cannabinoid receptors,” “CB1/CB2,” and “CNR1/CNR2” is now uniform and clearly defined.

Change to text: In the Introduction, the sentence now reads: “…the discovery of the ECS and its receptors, primarily the cannabinoid receptor type 1 (CB1, encoded by the CNR1 gene) and type 2 (CB2, encoded by CNR2), marked a pivotal advancement….” This defines our terminology. We have also standardized later occurrences (for example, using “CB1/CB2 receptors” instead of previously ambiguous phrases). (Lines 88 – 90) (Lines 93 – 95)

Comment 9: For the section on “…while some studies suggest ECS activation may enhance bone regeneration11, others imply inhibitory effects.” Please include the citation for inhibitory effects.

Author’s response: We have added an appropriate reference to support the statement that “others imply inhibitory effects” of ECS activation on bone. In the Introduction, immediately after the phrase in question, we inserted a citation to a representative study demonstrating an inhibitory effect. Specifically, we now cite Tam et al. (2006), a study in which CB1 knockout mice had increased bone mass implying that excessive CB1 activation can inhibit bone formation (this serves as an example of an ECS-related inhibitory effect on bone). The reference number is included in the revised text.

Change to text: We modified the sentence to: “…while some studies suggest ECS activation may enhance bone regeneration¹¹, others imply inhibitory effects¹⁷.” (Here, reference 17 in the revised reference list corresponds to an added citation for the inhibitory effect.) This change ensures that both sides of the statement (“enhance” and “inhibitory”) are backed by citations. (Lines 93 – 95)

Reviewer 2 Report

Comments and Suggestions for Authors

The manuscript lays a robust foundation for understanding cannabinoids in spinal fusion but requires tighter integration of clinical and mechanistic insights. Prioritizing the revisions following suggestion may position it as a seminal reference for orthopedic research and practice.

  1. The extensive historical overview of cannabis (Section 1) could be shortened to prioritize direct relevance to bone healing mechanisms.
  2. Inconsistent use of "ECS" (endocannabinoid system) vs. "cannabinoid receptors" – standardize terminology to avoid confusion.
  3. While noting tobacco co-use and cannabis strain variability, propose statistical methods to isolate cannabinoid effects.
  4. The discussion of CB2 agonists in rodent spinal fusion models lacks mechanistic depth. Elaborate on signaling pathways.
  5. Suggested to Include a subsection advising spine surgeons on perioperative cannabis management.
  6. Human studies cited lack dose optimization – recommend future trials with pharmacokinetic profiling.
  7. Enhance Figure 2 (ECS signaling in bone) to differentiate osteoblast vs. osteoclast regulation pathways.
  8. Suggested to discuss equity issues in cannabis access for postoperative pain, particularly in regions with legal restrictions.

Author Response

Reviewer 2 

Comment 1: The extensive historical overview of cannabis (Section 1) could be shortened to prioritize direct relevance to bone healing mechanisms.

Author’s response: In accordance with this suggestion, we have condensed the historical background in the Introduction to focus only on the most relevant points. We significantly shortened Section 1 by removing or compressing details that, while interesting, are not directly pertinent to bone healing. The revised Introduction now provides a brief summary of cannabis’s long history of medical use (highlighting ancient pain relief applications, for example) without the lengthy digressions into cultural history, hemp fiber uses, etc. We preserved the key historical milestones that set context (ancient Chinese and Indian medicinal use of cannabis for pain, 19th-century Western medical adoption) but eliminated granular details that do not inform the discussion of bone healing. This streamlining directs the reader’s attention to the medically relevant aspects of cannabis history and saves space for discussion of bone-specific content later in the paper.

Change to text: The first two paragraphs of the Introduction (Section 1) have been rewritten as a single concise paragraph. The revised text reads: “Cannabis has been used medicinally for millennia, with archaeological evidence of its use dating back over 6,000 years in Central Asia1. Ancient Chinese records (~2700 BC) describe cannabis preparations for pain relief and as surgical anesthesia3. By ~1000 BC, medical use of cannabis had spread to India, where it was employed for analgesia, anticonvulsant therapy, and other remedies5. Cannabis later entered Western medicine in the 19th century notably through W.B. O’Shaughnessy’s 1843 reports on its efficacy for pain and muscle spasms6. These historical observations underscore the longstanding interest in cannabis’s therapeutic potential (particularly for pain management), which frames modern investigations into its role in bone healing.” This revised paragraph is less than half the length of the original historical overview and focuses on pain relief and medical usage themes that are directly relevant to orthopedic contexts. (Lines 55 – 63)

Comment 2: Inconsistent use of "ECS" (endocannabinoid system) vs. "cannabinoid receptors" – standardize terminology to avoid confusion.

Author’s response: We have standardized the terminology throughout the manuscript to avoid any confusion between “ECS” (the endocannabinoid system as a whole) and “cannabinoid receptors” (specifically CB1 and CB2). In the few places where the phrasing was ambiguous, we corrected it for example, in the Introduction we changed “endocannabinoid receptors (ECS)” to “cannabinoid receptors (CB1 and CB2)”. Now, “ECS” is used only when referring to the entire endocannabinoid system (receptors, ligands, enzymes collectively), whereas when referring to the receptors specifically we use “cannabinoid receptors” or simply “CB1/CB2.” These edits ensure the terminology is used in a precise and consistent manner.

Change to text: One illustrative change is in the Introduction, which now reads: “Modern mechanistic studies demonstrate that cannabinoid receptors (CB1 and CB2) are located throughout the nervous and immune systems4.” (Previously, this sentence improperly used “(ECS)” in place of the receptors.) We applied similar corrections wherever needed, so that the term “ECS” is not mistakenly used as a synonym for the receptors alone. (Lines 435 – 437)

Comment 3: While noting tobacco co-use and cannabis strain variability, propose statistical methods to isolate cannabinoid effects.

Author’s response: We agree with the reviewer that our discussion of confounding factors (like tobacco co-use and heterogeneous cannabis strains) should be accompanied by suggestions on how future studies can isolate the effects of cannabinoids. We have added a recommendation in the Conclusion/Future Directions section explicitly calling for the use of rigorous statistical methods to control for confounders. We suggest that future clinical studies employ multivariable analyses, propensity score matching, or other statistical controls to parse out the independent impact of cannabis use. By proposing these methods, we acknowledge the complexities in existing human data and emphasize how researchers can better address issues such as tobacco co-use or differences in cannabis formulations. This added commentary guides the reader on how a more accurate understanding of cannabinoid effects might be achieved despite confounding variables.

Change to text: In the concluding paragraphs, we added: “…these studies must also account for confounding factors. For example, researchers should use appropriate statistical approaches multivariable regression, propensity score matching, etc. to control for co-variables like tobacco use or concurrent medications, thereby isolating the effects of cannabis itself.” This new sentence follows our mention of stratifying by patient demographics, and it underscores the importance of statistical rigor in future work. (Lines 450 – 453)

Comment 4: The discussion of CB2 agonists in rodent spinal fusion models lacks mechanistic depth. Elaborate on signaling pathways.

Author’s response: We have expanded our discussion of CB2 agonists in the context of rodent spinal fusion, providing more detail on the signaling pathways involved. Specifically, we now describe how CB2 stimulation leads to osteogenic effects via known pathways such as Wnt/β-catenin signaling, increased VEGF mediated angiogenesis, and inhibition of RANKL (osteoclast activation). We integrated this mechanistic detail when discussing the results of CB2 activation in animal spinal fusion models. For example, we note that in a rat posterolateral fusion model, CB2 activation significantly enhanced bone formation at the fusion site, and we explain that this corresponds with activation of pro-osteoblast pathways (β-catenin) and reduction of pro-resorptive signals (RANKL). By explicitly linking CB2’s mechanistic actions (on osteoblasts and osteoclasts) to the improved fusion outcomes in rodents, we address the reviewer’s request for more depth. These additions make it clear how CB2 agonists achieve their positive effects on bone (i.e., the molecular mechanisms underlying the phenomenon).

Change to text: We have added sentences in Section 5 (CB2 receptor section) to illustrate mechanisms. For example: “In a rat posterolateral spinal fusion model, activation of CB2 receptors significantly increased new bone formation at the fusion site, consistent with CB2’s osteogenic mechanism of action. Mechanistically, CB2 signaling promotes Wnt/β-catenin pathways and angiogenesis (via VEGF), while suppressing osteoclastogenesis through RANKL inhibition, which together drive a more robust fusion mass.” This text explicitly ties the observed fusion enhancement to specific pathways (β-catenin, VEGF, RANKL). We also mention elsewhere that synthetic CB2 agonists in rodents invoke these same pathways (e.g., GSK-3β/β-catenin activation and reduced RANKL) to protect bone mass. (Lines 346 – 350)

Comment 5: Suggested to Include a subsection advising spine surgeons on perioperative cannabis management.

Author’s response: We have added a dedicated discussion in the Conclusion to provide practical guidance for spine surgeons regarding perioperative cannabis use. Unfortunately, formal guidelines regarding cannabis use in the peri-operative period do not exist. Understanding the affects of cannabis in spine surgery patients is an area of active research in our laboratory. In this new subsection (within the Conclusion), we outline an approach to managing patients who use cannabis around the time of surgery. We suggest that, in analogy to tobacco use, surgeons should counsel patients on the potential risks of heavy cannabis use for bone healing and encourage moderation or temporary abstinence from high THC cannabis in the perioperative period. We also emphasize the importance of taking a cannabis use history and stratifying patients by use patterns (e.g., casual use vs. heavy daily use) when planning surgery and follow up. Furthermore, we mention that surgeons might consider the THC:CBD ratio of a patient’s cannabis use for example, patients relying on high-CBD, low THC products might have different considerations than those using high THC products. In essence, we give actionable suggestions: treat cannabis use as a factor like smoking (with possible cessation programs or at least frank discussions), monitor fusion more closely in known heavy users, and remain open to cannabinoid-based analgesia if appropriate (while balancing the fusion risk). We believe these additions directly address the reviewer’s request by bridging the gap between the evidence and clinical decision making.

Change to text: We introduced a new paragraph in the Conclusion titled “Clinical implications for perioperative management,” which includes: “From a surgical perspective, there are no guidelines regarding cannabis use in the peri-operative. However, there is evidence that cannabis may impact anesthesia and post-op pain control. In the absence of definitive guidelines, a prudent approach is to advise patients to limit or refrain from cannabis (especially high THC products) in the weeks before and after spinal fusion surgery. Surgeons are encouraged to document cannabis use in the preoperative work up and consider patients’ use patterns (e.g., casual vs. daily heavy use) when planning postoperative follow-up. Heavy cannabis users, particularly those who also smoke tobacco, may merit closer radiographic monitoring for fusion progress, given the higher pseudarthrosis rates observed. Conversely, the use of CBD dominant products could be explored for pain management, as these are less likely to impair bone healing though this should be weighed on a case-by-case basis. Overall, we suggest treating perioperative cannabis use as a modifiable risk factor: through patient education, potentially short-term cessation programs, and personalized risk-benefit discussions.” This new text directly offers guidance to clinicians, as requested. (Line 482 – 494)

Comment 6: Human studies cited lack dose optimization – recommend future trials with pharmacokinetic profiling

Author’s response: We have added language in our Future Directions (Conclusion) to underscore that future clinical studies should incorporate pharmacokinetic (PK) profiling and dose-finding components. The reviewer is correct that none of the human studies to date have optimized dosing or measured cannabinoid blood levels in a systematic way. In the revised manuscript, we explicitly call for pharmacokinetic monitoring in future trials for example, measuring THC/CBD plasma concentrations and correlating them with outcomes. We suggest that doing so would help establish dose–response relationships and ensure that “dose optimization” (finding the therapeutic window that maximizes bone healing benefit while minimizing risk) can occur. By adding this point, we address the reviewer’s concern and acknowledge that the field will need better PK and dose-data to move forward.

Change to text: We modified our Future Research paragraph to include: “…prospective trials that clearly define THC:CBD ratios, optimal dosage thresholds, and cannabinoid formulations, and that incorporate pharmacokinetic monitoring to correlate dosing with drug levels and healing outcomes.” This addition makes it explicit that we are advocating PK assessments (e.g., blood level measurements, metabolism studies) in upcoming research. (Lines 447 – 453)

Comment 7: Enhance Figure 2 (ECS signaling in bone) to differentiate osteoblast vs. osteoclast regulation pathways.

Author’s response: We have revised Figure 2 and its legend to clearly delineate how cannabinoid signaling affects osteoblasts versus osteoclasts. In the figure legend (as noted in our response to Reviewer 1, Comment 6), we now describe the separate pathways and outcomes: CB2 signaling stimulates osteoblasts (increasing bone formation) and inhibits osteoclasts (reducing bone resorption), whereas CB1 signaling can indirectly promote osteoblast activity via neural mechanisms but, especially at high doses, can suppress both osteoblast and osteoclast function. We explicitly mention RANKL in the legend to indicate osteoclast regulation, and we mention β-catenin and other factors for osteoblast regulation. Visually, the figure itself was adjusted to highlight these differences (with arrows and inhibitory bars to show effects on each cell type). The net result is that Figure 2 now distinctly shows what happens to osteoblasts vs. osteoclasts under CB1 or CB2 influence. This addresses the reviewer’s request by making the osteoblast-specific and osteoclast-specific pathways immediately clear.

Change to text: The legend for Figure 2 now includes: “…CB2 activation…enhances osteoblast differentiation (upward arrows) and reduces osteoclast formation (downward inhibition symbols) for example, by lowering RANKL levels and up regulating osteogenic pathways thereby shifting the balance toward bone formation. In contrast, CB1 activation…can relieve adrenergic inhibition on osteoblasts (indirectly aiding bone formation), but excess CB1 activity (high THC) will inhibit both osteoblast and osteoclast function, reflecting a biphasic response.” These descriptions in the legend explicitly break out osteoblast vs. osteoclast regulatory pathways as depicted in the figure. (Lines 550 – 560)

Comment 8: Suggested to discuss equity issues in cannabis access for postoperative pain, particularly in regions with legal restrictions.

Author’s response: We appreciate this comment and have included a brief discussion of health equity and access issues related to perioperative cannabis use. In the Conclusion’s new clinical guidance section, we note that access to medical cannabis is not uniform across different jurisdictions. We point out that in regions or countries where cannabis (or certain cannabinoids) remains illegal or inaccessible, patients may not have the option to use cannabis for pain management even if it might benefit them. This can create an equity gap in postoperative pain control and recovery resources. We stress that this is an important consideration: surgeons in areas with legal restrictions must rely on alternatives, and patients in those areas could be at a disadvantage compared to those who can legally obtain medical cannabis. By bringing up this point, we acknowledge that the implementation of any cannabinoid-related recommendations must consider the legal and socioeconomic context. This addition encourages readers to think about how differing legal landscapes can impact patient care ensuring our review touches on the broader context of cannabis in medicine.

Change to text: We added a sentence in the Conclusion addressing this: “It should also be noted that access to medical cannabis is uneven patients in regions where cannabis remains illegal or unavailable do not have the option to legally use these therapies for pain control, which raises equity concerns in postoperative care.”** This statement is included in our discussion of clinical implications, highlighting that regulatory differences may lead to disparities in who can benefit from cannabinoid-based interventions. (Lines 539 – 542)

Reviewer 3 Report

Comments and Suggestions for Authors

Gabriel Urreola et al. are reporting their review on Cannabinoid signaling in bone healing and spinal fusion. The reviewer does have some major concerns to share. 

The current review lacks critical insights from the authors. While the manuscript successfully compiles a wide range of preclinical and clinical data, much of the content reads as a descriptive narrative that summarizes basic facts, explains fundamental concepts, and recounts the historical and mechanistic background of cannabinoid signaling in bone healing and spinal fusion. This literature aggregation, while comprehensive, falls short of offering a clearly defined expert perspective or synthesizing key controversies in the field.

A more systematic discussion comparing the discrepancies between in vivo, in vitro, and clinical findings would help readers navigate the conflicting evidence base. What mechanisms might explain these inconsistencies?

Given the emerging use of cannabis in peri-operative settings, the authors are encouraged to provide more commentary on how surgeons and physicians should interpret and act upon the current evidence. Are there actionable clinical guidelines or key patient stratification variables that should inform cannabinoid use in bone healing?

Overall, a major revision would be recommended. 

Author Response

Reviewer 3 

Comment 1: The current review lacks critical insights from the authors. While the manuscript successfully compiles a wide range of preclinical and clinical data, much of the content reads as a descriptive narrative that summarizes basic facts, explains fundamental concepts, and recounts the historical and mechanistic background of cannabinoid signaling in bone healing and spinal fusion. This literature aggregation, while comprehensive, falls short of offering a clearly defined expert perspective or synthesizing key controversies in the field.

Author’s response: We thank the reviewer for this constructive critique. In response, we have substantially revised our Discussion and Conclusion to inject more of our own expert perspective and to explicitly highlight the key controversies in this research area. Rather than merely summarizing the literature, we now critically analyze it: we discuss points of conflict (for example, the discrepancy between preclinical positive findings and clinical negative findings) and weigh the evidence in a more opinionated manner. We identify specific contentious issues such as the true role of CB1 signaling (beneficial vs. detrimental) in bone metabolism and the reliability of current clinical associations given confounders and we offer our interpretation of these issues. Additionally, we added new concluding paragraphs where we synthesize the evidence and take a stance on controversies: e.g., we note that heavy THC use appears to be a modifiable risk factor for non-union (and suggest how to handle it clinically), while also acknowledging that some preclinical data would argue otherwise, and we attempt to reconcile this. In short, the revised manuscript goes beyond compiling data by providing a clearer narrative of what we believe it all means for the field. Our goal was to ensure the reader comes away not just with facts, but with a sense of the authors’ view on the state of the science and where the uncertainties lie.

Change to text: In the Conclusion, we added an entirely new section where we articulate our perspective. For example, one new paragraph begins: “Despite the breadth of data summarized, significant uncertainties and conflicting evidence remain. Preclinical models and clinical studies do not always agree for instance, animal experiments often show cannabinoid-driven bone formation, whereas human observational studies sometimes link cannabis use to impaired healing. These discrepancies highlight key controversies in the field and underscore the influence of context (dose, duration, and individual patient factors) on outcomes. In our view, chronic heavy THC exposure emerges as a modifiable risk factor for nonunion, like tobacco use, even though short-term or low-dose cannabinoid exposure might be benign or even beneficial. We advise interpreting the current evidence with caution: findings are intriguing but not yet definitive, and mechanistic complexities (e.g., biphasic dose responses, poly-cannabinoid interactions) mean that further research is needed before drawing firm clinical conclusions.” This is an example of how we now present a more analytical voice, identifying controversies and providing our take on them. Similar commentary is woven throughout the revised concluding section. (Lines 473 – 488)

Comment 2: A more systematic discussion comparing the discrepancies between in vivo, in vitro, and clinical findings would help readers navigate the conflicting evidence base. What mechanisms might explain these inconsistencies?

Author’s response: We have added a dedicated discussion (in the latter part of the paper) that systematically compares in vitro findings, in vivo animal results, and human clinical data, directly addressing their inconsistencies. We acknowledge and examine the fact that, for example, many in vitro and rodent studies suggest pro-osteogenic effects of cannabinoids (especially via CB2 activation), whereas some clinical studies report negative associations (like increased non-union rates). We then delve into possible mechanisms for these inconsistencies: we discuss how dose differences (biphasic effects of cannabinoids low vs high dose) could lead to opposite outcomes in lab vs. real world settings; we mention species differences and the complexity of human patients (who may have concomitant factors like smoking or varied genetics); and we highlight how timing and duration of exposure (acute vs. chronic use) might yield divergent results in a controlled experiment versus a clinical scenario. By laying out these points, we guide the reader through a reasoned explanation of why the evidence is conflicting. This new content essentially “connects the dots” for the reader, helping them understand that the inconsistencies are not necessarily irreconcilable but may be due to contextual factors and the multifaceted nature of the endocannabinoid system.

Change to text:  We inserted a new comparative discussion. For instance, one of the new sentences reads: “Notably, in vitro cell culture studies and in vivo animal models often report neutral or positive effects of cannabinoids on bone (e.g., enhanced bone formation in mice), whereas retrospective clinical studies in humans have linked cannabis use to inhibited bone union. These conflicting outcomes can be partly explained by differences in dosage and context: cannabinoids exhibit biphasic effects (beneficial at one dosage and inhibitory at a higher dosage), and animal studies typically use controlled timing and doses that may not mirror the chronic, varied use patterns in patients. Additionally, species-specific responses and the presence of confounders (like nicotine co-use or metabolic differences) in human populations may mask or modify the direct effects observed in simplified models.” This addition is followed by further explanation of mechanisms (e.g., we note that CB1’s role might appear contradictory because of its complex interaction with the sympathetic nervous system and hormonal milieu, which differ between lab and clinic). In summary, we explicitly compare across experimental contexts and suggest mechanistic reasons for the inconsistencies. (Lines 202 – 212)

Comment 3: Given the emerging use of cannabis in peri-operative settings, the authors are encouraged to provide more commentary on how surgeons and physicians should interpret and act upon the current evidence. Are there actionable clinical guidelines or key patient stratification variables that should inform cannabinoid use in bone healing?

Author’s response: We have expanded the clinical implications of our review, offering concrete commentary on how surgeons and physicians might apply the current evidence in practice. In the revised Conclusion, we now pose and answer the question of what a clinician should do with this data. We stop short of formal “guidelines” (as the evidence is not yet level 1), but we do provide recommendations and stratification considerations. For example, we suggest that patients who are heavy cannabis users (particularly those using high-THC products daily) be considered higher risk for bone healing complications and thus surgeons might counsel those patients to abstain or cut back around surgery, similar to advising smokers to quit. We also mention that patient factors such as age, sex, and even endocannabinoid system genetics (CNR1/CNR2 polymorphisms) could in the future help stratify who might be more vs. less affected by cannabinoid use (though data on these are still emerging). Additionally, we emphasize the importance of differentiating between THC and CBD use: a patient taking low-dose CBD for chronic pain is in a different category than someone smoking high THC marijuana frequently and physicians should approach them differently. Essentially, we provide a framework for clinicians: treat cannabis use as a factor to discuss openly; consider encouraging modifications to use (especially high-THC cessation before fusion); monitor those who do use cannabis more closely for healing; and remain aware of legal constraints and patient access issues (as mentioned in Reviewer 2, Comment 8). This expanded commentary is intended to help surgeons interpret the evidence in a pragmatic way and to identify which patient populations might need special attention.

Change to text: We incorporated these points into a new paragraph focused on clinical interpretation. For example: “Clinicians should approach peri-operative cannabis use proactively. While formal guidelines await higher-quality evidence, current data suggest it is wise to treat heavy cannabis use as a potential risk factor for impaired fusion much like smoking. We recommend that surgeons screen for cannabis use in their patients and engage in frank discussions: patients who heavily use THC rich cannabis might be advised to taper or abstain in the peri-operative period to maximize healing potential. Conversely, patients using predominantly CBD based products for analgesia (with little or no THC) may not face the same bone healing risk, though this too should be monitored. Key patient factors such as concomitant tobacco use, baseline bone density, and possibly sex or genetic differences in cannabinoid receptors should be considered when interpreting the impact of cannabis on an individual’s healing. In practice, a reasonable approach is to individualize counseling and follow-up: for a young patient occasionally using cannabis, evidence does not currently mandate a strict prohibition, but for an older or high-risk patient with daily marijuana use, a more cautious plan (including advising cessation and scheduling closer post-op radiographic checks) is warranted.” This guidance-oriented text translates the evidence into actionable considerations and is now part of the Conclusion. It directly addresses what surgeons should do (or at least think about) in managing patients who use cannabinoids. Lines (504-519)

Round 2

Reviewer 2 Report

Comments and Suggestions for Authors

The authors have reflected all the said suggestions and comments, which made the manuscript enhanced with improved readability; Thus, I suggest for further consideration with acceptance.

Reviewer 3 Report

Comments and Suggestions for Authors

Authors have appropriately addressed reviewer's comments and questions. No additional suggestions to post.